# Formation of Feminine Truth in Poststructuralism

**Abey Koshy**

Department of Philosophy, Sree Sankaracharya University of Sanskrit, Kochi 683572, Kerala, India;
abeykoshy@ssus.ac.in

**Abstract:** This essay traces the origin of feminine thought in poststructuralism, which opens up new vistas of experience that differ from traditional philosophical thinking based on a conceptual grasp of the world. Rather than viewing the feminine as the essence of the woman gender, it is seen here as the experience of a plurality of truths produced in the affectedness of the human body by the world. The representative function of language and methodology in traditional philosophy cannot capture the plurality of truths. Feminine experience is not a prerogative of women philosophers or feminist writers. It is accessible even to male philosophers. Since it is the outcome of the affectedness of the body by phenomena, it is accessible to all human beings, irrespective of their gender identities. The construction of the truth of entities in terms of their universal essence has a significant role in forming masculine and feminine experiences. Masculine experience is produced by the representation of conceptual truth by the self. Feminine is a kind of existence prior to self-formation that is in operation in all humans. The linguistic turn in philosophy created by Nietzsche and Saussure is the main force behind the growth of feminine thinking in poststructuralism. It marks the end of the abstract, concept-based thinking of the masculine sort and the formation of the differential thought of the feminine.

**Keywords:** feminine; masculine; language; truth; plurality; body; affect



## 1. Problematising Feminine Truth

This essay investigates the origin of feminine experiences articulated in poststructuralism rather than describing the nature of traditional feminism, which is engaged in a struggle to gain political identity for women in the social domain. It differentiates the 'feminine' from the woman's being, as seen in traditional feminism. A thought that affirms the multiplicity of truths emerging from bodily affects[1] can only be considered a feminine thought in the proper sense. It is not the presentation of any feminist ideology that makes a thought feminine. The essay explains how a non-metaphysical way of thinking initiated by Nietzsche and Saussure in the early twentieth century was instrumental in the emergence of a feminine perspective of truth in poststructuralist tradition.

The poststructural turn in philosophy marks the end of thinking based on transcendental realities and concepts. There is a shift towards representing empirical realities and the bodily dimensions of existence, as expressed in the slogan "back to things themselves" by Husserl. Like Husserl, Nietzsche's thinking of becoming and Saussure's reflections on language have also paved the way for representing worldly experiences in philosophy, as noticed in the works of poststructuralists.

Nietzsche's criticism of transcendental realities took a position against constructing abstract truths. It challenged the foundations of conceptual thinking until it prevailed in philosophy. Saussure's disclosure of the world's reality as a construction by language has also led thinkers to formulate ways to transform the identity thinking of metaphysics, which is felt as masculine.

The poststructural turn introduced by Nietzsche and Saussure was a linguistic turn in thinking[2]. This study, however, is not to claim that these two thinkers turned thinking and writing into a feminine expression in language. These thinkers have not produced any feminist thinking or writing. Instead, it only explains how their presentation of the truth

of the world was influential in emerging a new way of articulating human experiences for later thinkers, including poststructural feminist writers. These later figures led to the subversion of the ontology of traditional philosophy. Derrida is a significant figure among them in philosophy. Post-structural feminist thinkers like Luce Irigaray, Julia Kristeva, and Helene Cixous are other noted figures who refashioned the mode of thought and writing. The poststructural feminists claim that their writing aims to develop a feminine linguistic expression [1] (pp. 191–202), which they call *Écriture feminine*[3]. Such an expression could not gain legitimacy in modern civilisation. What makes the writing feminine is not the feminist message it conveys or its female authorship. Instead, the *ecriture feminine* provides a distinct mode of truth that is absent in the propositional form of writings presented in the dominant discourses of our times. By the dominant discourses, I mean all those writings of the natural sciences, social sciences, humanities, and culture that adhere to the traditional criterion of truth set by metaphysical philosophy. Demonstrability, abstraction, conceptual grasp, and verifiability are the criteria set by traditional philosophy. It believes that truth exists there *a priori*, for the human intellect to grasp and represent in their discourses.

The masculine nature of traditional philosophy can be explained as an attempt to provide structured discourse on whatever subject matter it deals with. The structure is a construction, schematisation, ordering, and formation of certain definite truths. Traditional philosophy does not see truth as a construction of human reason. It has always been a prejudice that truth exists independently of human cognition. Therefore, they held the truth and reality of the world to be stable and believed in the potential of human intellect to represent them in language. Nietzsche and Saussure, however, have pointed out the chaotic nature of the world and its incapacity to represent its truth in language. Modern civilisation has built on the foundation of concepts and truths provided by logocentric metaphysics. These realities, truths, and concepts were mere constructs imposed on becoming[4]. Constructing a world of 'being' can be seen as masculine for two reasons. Firstly, sensuality, nature, body, becoming, and pleasures are often associated with a woman's nature, whereas law, order, stability, reason, truth, culture, and self are considered manly characteristics. These concepts and truths are created through abstract thinking. Traditional philosophical persuasion of truth based on logical reasoning and conceptual thinking represents only the male's perspective of life. The philosophers who imposed this structure are males, and therefore it is an imposition of a masculine perspective on the world. While recognising the abstract essence common to a class of objects only as truths, philosophy discards the changing nature of things, beauty, sensuality, and bodily experiences into a lesser realm of truth. As the realm of multiplicity, it cannot be expressed in the language of logically oriented discourses. Its impossibility to be encapsulated in the conceptual language of philosophy does not imply its non-existence. This realm is what is being sought by the language of poststructuralism. As the other of masculine discourses, we can characterise them as feminine expressions in language.

Language is employed in traditional thinking to serve the representative function of meaning. However, feminine writings consist of truths that do not adhere to those criteria. It does not serve the functions of representation, conceptual grasp, demonstrability, and verifiability. In it, truth has to do more with the bodily experiences it carries. From the perspective of poststructuralism, the human search for truth is guided by various interests. Some consist of the accumulation of wealth, and others are driven by the desire to control the world's flux. Natural science and social sciences serve this function. Through this activity, they succeed in producing a humanised world in which man's life is secure with wealth, comforts, and equipment made by technology. They serve humans to face the challenges to existence created by nature in the form of various calamities.

Besides the human interest in creating a stable, livable world, human life and understanding are guided by various needs to satisfy desires, passions, aesthetic intuitions, and bodily and libidinal experiences. As psychoanalysis-inspired poststructuralism says, undergoing libidinal-bodily experiences produces fulfilment in life that creates an affirmative existence[5]. A drive to disrupt orderly, masculine, linear discourses is present in feminine

writings. It affirms the multiplicity of truths and affects produced by bodily contact with phenomena. Such writings need not necessarily be the contribution of female writers. The libidinal, bodily desire, which consists of experiencing the truths of objects as plural, is also present in certain works of male writers[6].

What is the source of a feminine language? The origin of language lies in human interaction with the world. Metaphysical philosophies see it as an interaction of the human mind with the world. However, from the perspectives of phenomenology and poststructuralism, human interaction with the world primarily happens at the bodily level. External objects make impingements on the body that produce affects. This is a different type of 'signification process' that comes before they are referred to the intellect. This interaction is pre-reflective and happens prior to organising affects into discrete concepts. There does not exist any man or woman at the level of bodily affects. Only drives, desires, and intensities[7] prevail there. The self-identity of a person disappears in a state of affectedness. The sensations of the sky, mountains, trees, animals, water bodies, plants, birds, flowers, and their fragrances affect the human body by which 'bodily intensities' are produced [2].

Concepts are formed only later, after they are referred to the brain. The brain classifies these experiences to form 'signs'. Words of language are such signs. Words are always the outcome of generalisation. At the level of word formation, the intensities are lost [3]. Therefore, instead of classifying them into discrete 'signs' through the application of reason, these intensities can be preserved in the body as intensities themselves. These intensities must be converted into new images and metaphors instead of generalising and reducing them into concepts that serve a mere 'representative' function of language. The intensities produced on the body by the affects of phenomena have to be reconfigured into images, sensations, and metaphors. Bodily affects and intensities are the experiences of multiplicity and becoming, which are the source of the feminine dimension of language. Intensities leave traces on the body as visual and auditory images. They are converted in art and literature into a different signification process, which Kristeva explains as semiotic expression [4].

Derrida refers to this as the 'feminine in operation' when he presents a deconstructive reading of Nietzsche's text *The Gay Science* [5]. Derrida demonstrates such an instance of the return of the feminine in a few of Nietzsche's stylistic practices. Deconstructing the passages of Nietzsche's *The Gay Science,* Derrida writes: "Nietzsche's writing . . .. Even if we do not venture so far to call it the feminine itself, is indeed the feminine operation." [5] (p. 57). The drive to develop an *ecriture feminine* by Irigaray and Kristeva comes from the understanding that meaning is an effect created by language.

This view does not subscribe to the stance that there are neutral, impersonal meanings and truths without any mediation of language. According to poststructuralism, no meaning exists outside of language. Therefore, if contemporary society is male-dominated, it is because of the masculine linguistic structure of the society. Any future reorganisation of the structure of society would not be possible without a reformulation of the nature of language. It requires finding a different linguistic expression. For such an expression to be feminine, a change in our attitude towards meaning is necessary. The meaning may not be understood in the model of mathematics and the natural sciences as deductive. The great value of meanings produced by signs, images, and sensations created in the affectedness of the body needs to be recognised. Poststructuralism broadly considers this as the domain of desire. Language in such contexts embodies desires[8]. Modern societies, however, have always looked at the domain of desire and its spillover into social life with suspicion. As a result, linguistic expressions formed from it are often dismissed as unwanted.

The manifestation of feminine experiences requires finding a new process of signification in language. The existing *langue*[9] of all modern world cultures is inhabited by signifiers produced from the masculine experience of the world. Saussure explains how the meaning, values, and practices of a culture or society are determined by *the langue* of that tradition [6] (pp. 20–24). Poststructuralism believes that the traditionally existing *langue*

has to be ruptured to formulate a new langue that illuminates the bodily, sensual, natural, and libidinal attributes of human existence that constitute the feminine in language.

Access to the feminine is not an exclusive prerogative of womankind. Even men can get to the feminine, provided they open themselves to affections that happen to their bodies from the external world. Instead of erecting an abstract conceptual order out of them, they have to accept the intensities produced by phenomena as truthful. For the entry into vast realms of experiences that hitherto lie submerged, neglected, and prohibited, rethinking the existing approach to language, meaning, and truth is required. True human liberation demands a new approach. Thus, coming towards the feminine dimension of one's existence shall be a concern for men as well.

The Saussurian linguistics that hold meaning as the result of the signification process assume importance at this point. To a large extent, the genesis of the feminine thinking of poststructuralism owes to revelations made by Saussure and Nietzsche on the actual role of language in human life. Nietzsche's reflection on language in the essay "Truth and Falsehood in an Extra Moral Sense" also questions the claims about the representative function of language. In the essay, he says that language cannot depict the truth of a world that is in flux. His insistence on building a life-affirmative philosophy prefers the truth of the body, sense, and material reality over the world represented by a subject through concepts, as it is rendered in traditional philosophies and natural sciences [7] (pp. 347–48). These reflections provided fuel for the emergence of the feminine thinking of poststructuralism.

The net result of this abstract thinking was the formation of a modern civilisation that does not approve of differences, multiplicities, bodily experiences, and sensual truths produced in human confrontation with the external world. The drive to categorise the world comes from an interest in dominating it because classifying the world's entities into genera and species helps men handle it according to their interests and conveniences. That makes it easier to exploit nature's resources to increase man's pleasure. For instance, the classification of the animal world into birds, reptiles, vertebrates, mammals, and insects is arbitrary. Man does it for the management of the organic world. In human interaction with the world, several new signs are being created.

Along with that, new percepts and meanings are also born. Signs represent a category of objects that are produced by homogenising them, discarding their multiplicities and uniqueness. The act of generalisation is the expression of a masculine drive[10] working in humans to dominate over other beings in the world. While classifying various types of organisms into insects, vertebrates, etc., based on some of their features, the singularity of each member of the class is denied. Organising entities in this way is an activity men undertake by imposing their perspective of truth on the world.

This is the drive working behind the masculine capitalist economy criticised by Luce Irigaray [8] (pp. 170–189). She explains its other pole, the feminine libidinal economy, as the attitude of letting things be as they are, led by preservative instinct. In place of forcible grabbing, the feminine libidinal economy is oriented towards giving and sacrificing oneself [9] (pp. 84–88). This attitude originates from receptivity and openness to the world, which is the outcome of the ability to be affected by objects and others at the bodily level. The feminine attitude manifested in experiencing the world and entities as objects that evoke sensations and affects will only be able to preserve nature and its resources. The masculine structure, however, is profit-oriented and negates what is natural and playful, leading to the development of a capitalist economy. Irigaray explains how the capitalist economy develops out of men's phallocentric desire to possess and control nature, as they possess women as objects of their sexual satisfaction [8]. On the other hand, the feminine libidinal economy is oriented towards giving, sacrificing, and fulfilling others over and above one's self-interests [10].

Natural and social sciences work on faith in the existence of an objectively real world before human interpretations. This trust was instrumental in giving legitimacy to those disciplines to explore and manipulate nature. It could get acceptance as the only right

dispenser of truth. It is believed that science alone possesses the correct method to investigate reality. The social sciences, which followed the natural sciences, adopted the same methodology in studying social phenomena. As a result, we naively accept the existence of a neutral reality outside our cognition of it.

## 2. Nietzsche's Opening to the Feminine

In the second half of the nineteenth century, Nietzsche, deviating from the long philosophical tradition, showed that the existence of an actual world, as claimed by metaphysics, is merely a fabrication intended to master and contain the changing nature of the world [7] (p. 330). Some see Nietzsche as a philosopher who created concepts like will-to-power, overman, and eternal recurrence. His interpreters tended to easily name him as a misogynist or male chauvinist based on his anti-woman pronouncements. If we move away from the peripheral layers of his anti-woman comments into the deeper layers of his thought, we find a pro-feminine stance in Nietzsche's writings[11]. He is the first to introduce a sexually different thought in philosophy. The sexual difference here is not merely a biological-gender difference between humans as man and woman. There is a sexual difference in human consciousness and thinking as well. Feminine consciousness is the result of the feminine experience of the world. His articulation of experiences produced by natural phenomena such as mountains, oceans, sky, dance, and music in philosophy were expressions of non-masculine experiences, which differed from the conceptual understanding of the 'being' of them in logocentric philosophy. In traditional philosophy, the truth of a mountain, rain, or tree is determined based on their universal abstract essence that lies transcendental to these objects. Therefore, the truth we get from the conceptual representation of phenomena could be seen as a masculine mode of truth. It is a masculine attitude to view the truth of entities based on their abstract essence. Studying a mountain based on its triangular form in geometry is such an abstract approach to a mountain.

The sexual difference he brought to thinking was not recognised until Derrida's reading in *Spurs: Nietzsche's styles* was published. His preference for thinking 'becoming' over being, beauty over logic, body over soul, emotion over reason, concrete reality over abstract concepts, particular over universal, and empirical over transcendental has marked a shift towards the feminine in thinking. Such a depiction of the truth of the world in Nietzsche's writing prompted Derrida to say that Nietzsche's philosophy has articulated a feminine truth of phenomena [5].

Nietzsche has created a turn in philosophy by inaugurating non-metaphysical thinking by rejecting its truth claims set on transcendental foundations. According to his doctrine of 'perspectivism'[12], the understanding of philosophers represented only one perspective of the world [7] (p. 305). Unfortunately, this understanding is the one incorporated by natural and social sciences.

Nietzsche exposed the male-centric characteristic of this perspective while saying that the creation of law, order, stability, reason, truth, culture, and self is a manly activity[13]. Philosophy has so far recognised only the generic essence common to a class of objects as truth, which is received through abstract thinking. Traditional philosophical persuasion of truth based on abstract reasoning and conceptual thinking represents the male's perspective because it intends to control and dominate the entities. The purpose of creating a world of being is to control sensuality, nature, body, becoming, and pleasures, often associated with a woman's nature. Nietzsche, deviating from transcendental questions of traditional metaphysics, has brought philosophy down to earth to explain sensual, bodily, and worldly truths. Since traditional philosophy associates body, sensuality, and worldly beauty with a woman's nature, Nietzsche's depiction of them in philosophy shall be seen as a stance favouring a feminine experience of truth. About the traditional philosophical view of truth, Nietzsche writes:

"without accepting the fictions of logic, without measuring reality against the purely invented world of the unconditional and self identical, without a constant falsification of the world by means of numbers man could not live—that renouncing false judgments

would mean renouncing life and a denial of life. To recognise untruth as a condition of life—that certainly means resisting accustomed value feelings" [11] (p. 12).

Nietzsche's criticism of truth, morality, religion, and metaphysics has to be considered a stance in favour of promoting the value of sense, appearance, beauty, and woman. For Nietzsche, life is seductive like a woman [12] (p. 272). While man, in Nietzsche's thought, is the figure of life negation, the woman is the figure of life affirmation. He equates life to a woman; for him, life's most powerful magic is its feminine character. He writes that the world "is covered by a veil interwoven with gold, a veil of beautiful possibilities, sparkling with promise, resistance, bashfulness, mockery, pity, and seduction. Yes, life is a woman" [12]).

For him, the untruth that women and art represent is more powerful and valuable for life than truth. A woman desires the art of grace and playfulness that man has lost. This untruth is also what men should seek. In the preface of *Beyond Good and Evil*, Nietzsche proposes that the 'woman' shall be the model for philosophical thinking. He asks how a dogmatic philosopher who is inexpert about women will be able to find her truth? [11] (p. 1). A woman's truth will not be revealed to a truth-seeker who searches with the methodology of abstract reasoning. If life and the world are like a woman, in place of abstract reasoning, humans shall be affected by them. Nietzsche depicts woman as untruth. Not only woman, but the entire cosmos is also untruthful because the truth is a fabrication or interpretation of the philosopher. Truth is non-existent. Disinterest in truth, he says, is the feature of feminine will: "she does not want truth: what truth to a woman? . . .her great art is lie, her highest concern is mere appearance and beauty" [11] (p. 163). As the figure of untruth, appearance, animalness, body, nature, and worldly beauty, the woman is more acceptable to Nietzsche than the man of reason. When man negates worldly life, the woman alone affirms worldly life. He says a woman's great art is dissimulation, simulacra, and untruth.

Notwithstanding the anti-woman comments in his philosophy, Nietzsche's preference for natural existence, the beauty of the world, and sensuality over abstract truths has to be seen as a feminine attitude, which was a counter position to the masculine mode of traditional philosophy[14]. For Nietzsche, the world is false, and there is no truth to find [7] (p. 550). The magical spell of appearance of the world, which he associates with woman, is what one ought to seek [12] (pp. 271–272). Nietzsche's philosophy, therefore, is a stance in favour of nature, appearance, sense, body, animality, natural living, and sensuality. Thus, it reflects a feminine perspective of life.

This view of truth proposed by Nietzsche was a stimulant for the feminist thinkers of poststructuralism to say that they disown the traditional perspective of truth and perceive the reality of the world and values differently. They ventured out to produce writings that enabled them to express their experience of reality. They call this the *ecriture feminine*, the feminine mode of writing. Nietzsche is observed to be producing such enigmatic feminine writing that consists of plural significations. Therefore, though Nietzsche holds an anti-feminist stance, his philosophy attempts to inscribe a feminine language, a feminine world perspective, and a feminine truth in philosophy [13] (pp. 116–129).

Nietzsche was the model for the poststructuralists who proceeded with the thought to exalt the value of human sense experience. The body is the ground of all sense experience. The body represents two meanings here. In one sense, the body symbolises the world and material reality that oppose transcendental truths. In another sense, the body is the signifier of all those elements that are cast out from the realm of pure truths, such as feelings, desires, passions, love, sex, beauty, and whatever affirms human earthly life. The body in contemporary thinking is set against the 'self', formulated by the metaphysicians as the essence of the human being. To Nietzsche, self (soul) is merely a layer of consciousness about laws, social norms, and memories that acts as the agency in the individual to prevent the expression of instincts and drives. It is constituted through the internalisation of instincts [14] (p. 84). Therefore, Nietzsche's affirmation of the body is also considered a beneficial project for affirming the feminine perspective of life [15].

Thus, the final consequence of logocentric thinking that discards the affective dimension of humankind is life denial. Nietzsche observed this attitude of traditional philosophy as a case of nihilism. He defines nihilism as positing transcendental truth above the world of change and senses, thus depreciating the natural world [7] (pp. 9–39). To increase spiritual power, traditional philosophers keep sensuality, beauty, and women away from their lives. Nietzsche perceives such an ascetic denial of life, body, and the world by philosophy and religion as the source of nihilism. Nihilism means denying worldly life, senses, body, and woman to secure morality and otherworldly life. Criticising nihilism, Nietzsche observes that the aversion for sensuality and worldly beauty is why many philosophers remain unmarried [14] (pp. 106–107). Man's pursuit of achieving power through religion and science could not have been possible without casting aside the body and its desires. The woman who represents the body is considered a hindrance to man's path to spiritual and scientific growth. Traditional religious and metaphysical discourses perceive women primarily as figures of passion, illogic, body, sex, etc. To gain power, man constructs himself as the self. For spiritual growth, he cast aside women from their personal lives [14] (p. 107).

On the other hand, man is considered to possess all positive qualities such as rationality, logic, spirit, mind, and similar qualities. It can be observed that Nietzsche's criticism of asceticism directly attacked religion, metaphysics, and the sciences, which were basically men's projects. While building modern civilisation based on these projects, men have forced women to conform to masculine norms and standards. Nevertheless, men were also cutting themselves from the world of the body and its desires by this act. Nietzsche considers this a strong case of life denial. His criticism of nihilism can be seen as a direct attack on the masculine perspective of the world. Nietzsche, who criticised nihilism as the ascetic denial of bodily desires, opened up a site for thinking of the feminine in philosophy.

Viewing worldly life as ephemeral, logocentric philosophy constructs a transcendental world above nature, which Nietzsche considers nihilistic. It is a man's project to categorise entities by generalisation based on some of the common observable features. It is intended to manage, dominate, and control entities. It is considered the expression of a masculine attitude. Thus, metaphysical thinking that creates a transcendental structure of truth over the empirical world is patriarchal. Therefore, Nietzsche's criticism of nihilism contains an indirect criticism of patriarchal structure. Patriarchy and metaphysics originate from the same source. Both create a structure above change to dominate over nature, women, and sensual realities.

Due to these views of Nietzsche, poststructural feminist thinkers find in him an ally to their project of criticising patriarchy. Though Nietzsche does not critique patriarchy in any place of his works, it is observed that his criticism of the nihilism of metaphysics was an indirect attack on patriarchy [15]. Debra Bergoffen perceives patriarchy as the child of metaphysics. Patriarchy and metaphysics play the same role when both create a hierarchical structure above life and the world intended to control change, pleasures of the body, women, sensations, and the beauty of nature. Some feminist critiques show that patriarchy is intertwined with the life-denying attitude of metaphysical thinking [8] (pp. 74–77). Logocentrism has been deeply in operation since metaphysics was accepted as the legitimate dispenser of truth. The logocentric drive associated with the masculine greed to relish worldly objects for pleasure led to the domination of the earth and woman's body. Derrida's use of the term phallogocentrism is intended to explain the link between the classification of the world in terms of logical categories and the motive of overpowering nature through that act.

In the psychoanalytic explanation, libidinal desire is expressed in the tendency to unite with other bodies to form symbiotic relations [16]. Masculine desire, represented by 'the law of the father', on the other hand, only prevents such formations and maintains individuals as separate subjectivities in the phallocentric symbolic order of society. However, as psychoanalysis reveals, the pre-oedipal phase of every human being consists of the libidinal relationship with the 'other', which is disrupted by the socio-symbolic law of the patriarchy. Thus, patriarchy prevents the joy of libidinal unity and seeks pleasure in the symbolic

power provided by the possession of objects. The source of pleasure in masculine nature lies in preventing libidinal desires in humans to exploit, possess, and accumulate the world's goods.

Life negation results from positing higher truths above life and the world [7] (p. 12). The higher truths are those ideals and principles we see in various human sciences and religious discourses whose origin, in Nietzsche's opinion, is in Christianity and metaphysical thinking.

For him, the drive for truth comes from the instinct to preserve the human species. The construction of truths was essential for making life manageable and tiding over the tragic consequences of change (becoming). In Nietzsche's words, in the ocean of 'change', to find stability in life, man imposes the character of 'being' upon becoming [7] (p. 330). This is achieved by schematising the world through categories such as 'substance', self, 'goodness', past, future, and so on. Though these categories provided stability for life, they turned into ultimate truths and standards for measuring reality in all realms of life. Consequent to the acceptance of 'being' alone as truth, whatever evades its totalising logic of identity and non-contradiction has been negated as false. Aesthetic joyfulness and knowledge produced in sensations and affects all fall under this category. These are truths that evade the grasp of conceptual thinking of traditional metaphysics. They are felt as plural experiences of truth.

Nietzsche's philosophy, thus, was instrumental for poststructuralists like Derrida to rethink the so-far-accepted views about truth, language, and human self-identity. Up until then, truth and self-consciousness were explained in an essentialist manner as pre-existing realities. Modern civilisation, resulting from the metaphysical thinking of the last two millennia, still thrives on believing in a world made of stable substances, essence, and selves. While interrogating modernity, Heidegger has also commented that metaphysical understanding of the truth of entities was an abstract understanding of entities based on reducing their common properties and similarities to create their 'being' [17]. Metaphysics could never recognise the truth of the embodied existence of entities. It is not the particularity of an entity but the similarities among entities that are put together to form a class or category. This is the ontic attitude of European metaphysics criticised by Heidegger. He has tried to replace it with a fundamental ontology. Derrida also criticises this tendency to create abstract concepts as a 'logocentric' bias of traditional Western philosophy.

## 3. Plurality as Feminine

Is it possible for humankind to represent the plurality of truths in language? It is the most important question posed by poststructuralism. Due to its failure to grasp multiplicity, traditional philosophy castigates plural experiences in the domain of otherness. Unable to recognise them as truths, traditional philosophy projects them as either mystical or aesthetic and pushes them into the domain of feeling or faith. However, metaphysics does not recognise that its failure to convey plurality is only a failure of the representative function of language and not a failure of language as such. To capture plurality, the potential of language has to be fully explored. The mission undertaken by poststructuralism in the philosophical tradition is to get over this shortcoming of metaphysics. Therefore, poststructuralism is a turn in philosophy that seeks to represent the so far unrepresented.

Traditional metaphysics fails to convey plurality because it is not ready to discard its already accepted position of viewing words as a vehicle for representing the abstract essence of things. Words are not static objects with a fixed meaning but are always in a state of change. In the poststructural perspective, writing is not simply a representation of pre-existing meaning but is the primary site where meaning is produced.

Poststructuralism is born out of the attempt to represent the realm of otherness, which was cast out by conventional philosophy as a set of experiences opposed to reason. Unless philosophy can convey, delineate, and express those experiences discarded as obscure and untruth, human beings will not be able to move to a higher plane of existence. The failure philosophy has faced in this regard in the last two millenniums has to be overcome. In Derrida's writings, an attempt is made to capture the plurality of truths by utilising the

fullest potential of language possible. He has shown in his writings how writing creates meaning through a process of deferring, whereby the meaning of a word is continuously postponed and deferred to other words, creating a never-ending chain of significations [18].

While metaphysics uses words to represent a category, it negates the specificities and particularities of each thing. All modern disciplines based on metaphysics have negated the plural nature of things in order to put them under universal categories of logical discourse. The universal is always what transcends the body and singularities. Such an approach is applicable only in scientific disciplines that are oriented towards augmenting material production.

Productivity, which brings forth new things from the raw natural order, is considered masculine. The ability to create new things out of wild nature is a peculiar capacity of human beings that differentiates them from other species. It led to man's domination over nature and other entities. Such nihilistic manifestations of reason enabled man to enter historical life, where new things and social forms are perpetually produced. If man had not transformed nature into culture, the human species would have remained in the natural order outside history. However, such a non-productive existence is looked down upon as passivity. The passivity that leaves the species in the natural order is criticised as a feminine attitude. Since it does not bring power to the species, masculine qualities of activity, vigour, and constructiveness have been given more value than the feminine. It led to a gradual undermining of the feminine.

Being a domain of change, uncertainty, and adversity, nature has been seen as an insecure place to live. In this regard, man's development of conceptual knowledge in various disciplines of the sciences may be seen as an attempt to avert disasters of nature. Writings that happen outside of conceptual thinking do not bring such results. The imaginative production of works of art such as poetry, painting, and music has also been associated with the feminine realm due to their lack of contribution to material benefit. Activities such as play, laughter, child rearing, and wandering, which do not produce material results, are thus looked down upon and treated as feminine acts.

However, the value of knowledge cannot be measured merely in terms of its utility for human survival. The creation of moments of intensity in life, a favourite project of the poststructuralists, happens in another way. What Nietzsche calls life enhancement is served by moving through intense experiences, which are the outcomes of the body's exposure to beautiful phenomena in the world. The 'signs' produced out of the bodily intensities need not be turned into concepts. Signs can evoke the flow of libidinal desires underlying the body, which Kristeva calls semiotic chora [4]. Signs can remain as visual and auditory images and metaphors, which can create multiple plays of signification. In them, knowledge is not formed through conceptualisation to serve various utilitarian interests of man, as seen in the natural and social sciences. On the contrary, knowledge is meant to create difference, play, and intensities in the body that take humans to a higher plane of existence.

Nietzsche's criticism of the life-denying character of the truth of traditional philosophy and the natural sciences indirectly criticises the masculine economy. Abstract knowledge rejects nature and bodily desires as inferior realities. Nietzsche's affirmation of the flux of the world is thus a feminine approach to life. With this stance, Nietzsche accepted the plural nature of truth and existence. It will be considered a more valuable option. The plurality of truths produced by sensations and affects must be activated and turned into multiple linguistic significations and thoughts. Such thoughts and writings are what we precisely call feminine.

Nietzsche's critique of the representative function of language was a crucial milestone in developing the feminine in writing. His linguistic critique of truth is delivered in the essay 'On Truth and Falsehood in an Extra Moral Sense", which was similar to the thought on language provided by Saussure. The central claim of it is that the logical axioms rest entirely on words made by man. Words, in turn, are nothing but metaphors and thus incapable of revealing the true character of the things signified by them. It is

argued that language, instead of giving us a true account of things in the world, is a referentially unreliable set of almost entirely arbitrary signs made by man to safeguard life and species [19] (pp. 45–46).

The main idea articulated in this essay on language can be summed up in a few sentences. Words do not designate things that are constantly changing. Words are metaphors for real things. Man cannot grasp the real nature of things through his sense organs. Rather than truth, the real value of language is a utilitarian one. It is meant to hide from humans the hostile and changing nature of the universe. It is employed to preserve humans from destruction. No knowledge of a world beyond our language is available to us. Between words and things, there is no direct relationship. Words are said to be the distant and distorted echoes of the real. Nietzsche argued that these echoes or rudimentary elements are given coherence according to rules entirely invented by man. Consequently, the relationship between words and the real world is metaphorical or aesthetic. The metaphysical perspective about man as a perceiving subject of an objective world is reinterpreted in Nietzsche's scheme as man as the creator of language. If language thus constantly produces only fiction, our cognitive apparatus can be seen as a falsifying mechanism. Therefore, the legitimate logical discourses of sciences, religion, and philosophy are unable to articulate or express the flux and plurality of life because, with words, we always convert things into substances having immutable forms. Logocentric means of expression are useless for expressing becoming.

The implication of the above criticism is twofold. Firstly, it suggests that all the legitimate mainstream accounts about the world, whether they are those of science, social science, or religion, are merely metaphorical and thus are interpretations of phenomena. As a result, the validity of differentiating scientific discourses from aesthetic ones based on greater truthfulness is challenged. If all truths are human attempts to create stable concepts by scheming chaos, the plural truths emerging from bodily affects and sensations are also to be counted as valuable. Poststructuralism thinks such truths are more valuable than logocentric discourses because they enhance the powers of the body by creating joy. However, logical discourse creates dullness in the body. Thus, interpreting the world in multiple ways is necessary to creatively reorganise life and society.

Secondly, it shows that life has no essential meaning, as conventional thinking claims. The meaning and value of life will be renewed as we evaluate them differently. Poststructural criticism of nihilistic metaphysics played a significant role in legitimising the expression of plurality in language. Like Nietzsche, it also assigns language and art a significant role in capturing the experiential pluralities of life. It is not only the creation of works of art; writing in general has to be turned into an embodiment of desires. This is the feminine dimension of discourses that poststructuralism wanted to activate, which have been silenced, condemned, and rejected as valueless by the male-dominated modern culture.

## 4. The Temporality of the Feminine

The experience of truth as plurality creates a different experience of time from the standard time set in the clock and calendar in modernity. Standard time is the product of linear history conceived by Enlightenment philosophies. From a poststructural viewpoint, the Western idea of history can be seen as a man's experience of time. For the Enlightenment thinkers, man is a species with historical consciousness that differentiates them from other organisms of nature. The experience of time as a linear development from the past to the future is the cause of historical consciousness. Hegel's explanation of world history as a teleological development adheres to this model. The teleological living of time is to be considered the expression of masculine desire for mastery over the world.

In poststructuralism, instead of progression, time is paradoxical and consists of constant ruptures and breaks. Instead of attempting to fix essence to time by dividing it into past, present, and future, poststructuralism perceives time as multiple and fragmented. For it, time is more or less repetitive than linear. Language and discourse play a crucial role in constructing the experience of time. Some of Derrida's writings that disrupt tra-

ditional linear narratives create an experience of time through repetition, fragmentation, and looping in his texts. It creates a sense of rhythm and repetitive temporality that is not strictly sequential.

Nietzsche's theory of 'eternal recurrence' may have functioned as a model for the poststructural conception of a repetitive temporality. Nietzsche presents eternal recurrence as the experience of eternity brought about by man's involvement in the momentary experiences of the 'present' tense. To live in the present, one has to eliminate the historical consciousness gained from metaphysics. The experience of time as a sequential passage from past to future is felt mainly due to the cognitive approach to the world. The cognitive grasp of phenomena happens through abstracting the essence of things from their various appearances in moments. Nietzsche viewed the world as in the process of becoming, continually in flux. Though they do not have any essence, in a bid to grasp and control objects, human beings impose permanent essence on things. Nietzsche observes this imposition of the character of being on becoming as the expression of the human nihilistic will to power to dominate over phenomena. An image of truth over the objects is stamped to keep the chaos under control. Self-formation in humans is also the outcome of structuring chaos. Such a self perceives the changes taking place around it in a sequential manner. Kant has already pointed out how sequential order in the perception of the world rests on the internal time consciousness of a rational self. Objectivity to time is thus the outcome of human activity to contain chaos by the rational self.

Poststructuralism does not see time as an objectively and independently existing phenomenon outside the human consciousness. Instead, time consciousness results from the various unconnected experiences a person undergoes in life. The human sensation of objects in the world creates intensities in the human body. These intensities' frequency varies depending on the type of experience one passes through. The intensity produced by a musical notation differs from the intensity created by a love experience. Likewise, immersing in the fragrance of flowers during the spring season would be different from the intensity produced during the departure of our beloved.

Julia Kristeva points out that woman, as a being whose connection with the world is different from that of men, experiences time differently. She argues that women's experience of time is shaped by their biological rhythm and psychic relationship to childbirth, nurturing, and caring for others. Unlike the man who produces new things by converting natural raw materials into cultural artefacts, the woman who is confined to the rhythm of nature experiences time repetitively [20] (pp. 191–192). *Ecriture feminine* consists of the articulation of the intensities produced in the body during its affectedness by seasons, rainbows, love, music, dance, and the body's biological rhythm.

Nietzsche's theory of eternal recurrence acted as a model for articulating the feminine experience of time in poststructuralism. Debra Bergoffen writes, "Nietzsche's call for trans-valuation must include a demand that feminine temporality be explored for possible antidotes to the nihilistic poison" [15] (p. 81). Repetition in eternal recurrence is a repetition of the moments that reject the past and future as artificial constructs of those who approach time ontologically.

Poststructural criticism of historical consciousness shall be seen as an attack on the male's temporality. As Nietzsche has shown, animals and children, in the absence of memory of the past, live happily in the present [21] (p. 61). In his opinion, it will be the model for the human being as well. On the other hand, the human tendency is to cling relentlessly to the memory of the past. Instead of cultivating a certain amount of forgetfulness, man idealises the past and ensures security in the future. Man is made into a labourer for the future. Masculine historicism equates happiness with achievements. The present is never considered to be perfect.

## 5. The Linguistic Challenge to Masculine Metaphysics

Saussure's structural linguistics has neither presented any feminine linguistic theory nor any critique of metaphysics. Being a linguist, he was not well exposed to metaphysical

debates about truth. Nevertheless, his revelation that truth is always mediated through language has fueled later thinkers to challenge metaphysical theories that stood for the unmediated existence of truth.

It was Saussure who showed that the reality of the world is not the reality of things in the world. According to him, humans cannot understand the truth of 'things' as they are. He believes we can only know things as they appear to the sense organs. In sense perception, instead of grasping the real nature of things, we perceive them merely as 'signs'. Furthermore, our access to things depends entirely on language [22] (pp. 65–66). A thing gains its meaning only with the formulation of a *sign* to refer to that thing. Instead of 'representing' the freely existing concepts or things, linguistic signs articulate and create concepts arbitrarily. Turning concepts into mere signs by Saussure's structuralism disturbed the foundation of Western logic, which believed in the existence of meanings and concepts independent of signs.

In the view of logocentric philosophy, the truth of a thing is understood based on its essence, which comes from the universal category to which it belongs. This approach is what Heidegger calls the 'ontic'[15] outlook of metaphysics. According to metaphysics, truth does not lie in a concretely existing thing but in its abstract essence. In metaphysical traditions, the essence is considered as realities existing timelessly, trans-culturally, and trans-linguistically. Metaphysics looks at language simply as a device invented by man to represent an already existing truth. However, with the help of Saussure's thought, we now see that the transcendental existence of meaning is merely a belief in metaphysics. The meaning, in fact, always differs from culture to culture and from time to time.

Saussure has exposed how the identity of phenomena is arbitrarily constituted in the signifying process of language through the denial of their difference. Meaning, he explained, is always determined simply based on the differential relation the 'signs' have with other signs or objects [22] (p. 117). It paved the way for the development of the differential thought of poststructuralism. The radical reflection of Saussure has provided poststructuralists with armaments for combating the essentialism of metaphysics.

They have highlighted the fragility of human claims about the ability of human thinking to represent the truth about the world. Since the meanings produced through signs are arbitrary, whatever discourses we make, whether of philosophy, natural science, religion, or art, none contain any absolute truths. Saussure's reflection helped later thinkers and writers use language to articulate becoming by utilising its potential to the fullest extent. Saussure had only set the groundwork for the practise of plurality in linguistic expression. The actual accomplishment of plural expression was undertaken only later by the poststructuralists.

Derrida's development of Saussure's structuralism into a poststructuralist theory of language was indeed a crucial milestone on that path. It helped to manifest *differences* in human thinking. Earlier philosophical discourses have entirely consisted of identity thinking, by which one could have represented only sameness, neglecting the differences the entities have with one another. The same was true of the attitude of metaphysical philosophy towards approaching ethical and cultural ideas. The concepts of good, evil, beauty, justice, crow, horse, men, etc., are a few examples. While identity is accepted as the criterion of truth, the difference between one horse and another was not represented. 'The law of identity' of logic has developed from this attitude. When two entities are not identical, they are placed at opposite poles. The second law of thought, 'the law of excluded middle,' for instance, denies the possibility of the truth of a thing that comes in between A and non-A. Likewise, in a system of colours, the value of the grey shade between black and white was not recognised. This is the ground for categorising all human actions as moral or immoral, just or unjust, and good or evil. No signs are available to designate moral actions that surpass the rules of conventional morality. This attitude gives room for human thinking to brand non-identical things as false. If an action is not considered good, it is evaluated as evil. The possibility of standing in a space between truth and falsehood is denied. This is what the law of the excluded middle does. Traditional philosophy's

trust in the language's ability to represent reality rests on the laws of thought of logic. According to this philosophy, a thing has to have a permanent identity. As a result, the law of identity and the law of non-contradiction do not allow us to understand the differential dimensions of things. For traditional philosophy, an identical thing alone guarantees linguistic representation. Against this stance, poststructuralism aims to illuminate the grey area between truth and falsehood, or black and white, which has been left unarticulated in the logocentric discourses.

It enabled Derrida's deconstruction to legitimise all kinds of linguistic practices on par with any discourse in any field. It could not have been possible without Saussure's perspective of language as an arbitrary differential system of signs.

However, how is language able to articulate chaos? This is the highest question raised by Derrida's deconstruction. In some of his deconstructive writings, Derrida has turned language into a plural play of signification that would express the inexpressible. Nevertheless, traditional discourses mostly convert the world into familiar and discrete things instead of reflecting chaos. It enabled them to hide chaos. It projects the intelligible alone as truth. In its bid to totalise them, metaphysics reduces the plural nature of phenomena while converting them into intelligible objects. Thus, the indeterminate nature of the world is transformed into determinability. Linguistic signs usually cannot designate the singularity of things in our momentary experiences. On most occasions, signs stand for transcendental, abstract, universal representations of a category, formed from the generalisation of several particular experiences of the phenomenon.

Why do we want to generalise our experiences? Can we not accept, as Heidegger argues, our particular temporal experiences as a revelation of the truth of the phenomena? [23]. Why do we always try to abstract our experiences? This drive for abstraction is a masculine approach to knowledge. Saussure's linguistic theory can be used to explain how a masculine culture comes into being through arbitrary categorisation and abbreviation of the meaning of things through 'signs'. Without categorisation, the world will be felt as indefinite and chaotic. It is the language that helps man give structure to the world by creating concepts.

Nietzsche perceives this drive as a desire for mastery over the world, which he explains through the idea of 'the will to power'. Some humans want to make the world a hierarchically ordered one. This tendency comes from the drive to have a solid hold on the phenomena surrounding us. They want to make our surroundings predictable instead of leaving them as strange, unknowable phenomena. In Nietzsche's opinion, the fear of indulging in plural, momentary experiences makes humans logically structure the world. Its masculine tendency lies in its desire to have control over things. The feminine attitude of the species is expressed in the acceptance of fleeting experiences, which is part of the flux of nature. The language that articulates becoming makes the writing a feminine one, which is called *ecriture feminine* by French feminists.

A subtle examination of Saussure's theory would help us understand how a male perspective operates behind the human drive to see the world as something composed of stable entities and concepts. There is a history of the development of the masculine in thinking. It coincides with the emergence of logocentric metaphysics, whose first form Nietzsche locates in the dialectical thinking of Socrates. According to him, the world-denying spirit of nihilism begins with it [24] (pp. 89–98). The coincidence of logocentrism's advent and nihilism's development is not accidental.

Instead of understanding patriarchy as overpowering women by men, it should be seen as the structure-making drive of the masculine attitude. It comes into being through a reductionistic interpretation of the meaning of phenomena when placing them under specific universal abstract categories. These abstract categories are what Saussure identifies as signs.

Nevertheless, when Saussure explains the arbitrary function of signs in the constitution of meaning, he leaves room for a different reorganisation of the meaning of phenomena. While perceiving 'meaning' merely as a result of the differential play of signifiers, poststruc-

turalism, inspired by Derrida, has gone a step ahead of Saussure. In its bid to articulate the in-articulable, to express the inexpressible, poststructural writings turned signs into traces. If the meaning of a sign is arbitrarily constituted based on the agreement of the people living in a tradition, its meaning can differ from context to context. To Derrida, meaning is a differential play produced by the signs in various contexts of reading a text [25] (p. 280). It is shifted from time to time depending on the occasions and contexts of the reading activity. These readings lead to the production of new writings and texts that alter the existing experiences of truth. In such texts, the signs function as *traces* that turn signification into infinite play. In Derrida's opinion, writing enables the articulation of plural truths that emerged in human contact with phenomena that remain unrepresented in 'logocentric' language.

The most radical consequence of such a signifying process is its disruption of the existing structure approved by societies as truths and norms for their people to follow. Saussure speaks of *langue* which is the underlying linguistic structure of a society by which the members of that society can speak and act in meaningful ways [22] (p. 15).

When Jacques Lacan employed Saussure's notion of langue to explain the structure of humans' unconscious, he also recognised the arbitrary nature of human nature, values, and truths. The unconscious of individuals varies from tradition to tradition based on the structure of the *langue* of the linguistic community. Since words, signs, and concepts are regulated from a masculine perspective, a patriarchally structured unconscious is formed in all modern communities. Lacan explains how 'the unconscious' is constituted by the patriarchal laws and meanings of the society, which are assimilated by every child as soon as he/she enters the socio-symbolic order necessitated by castration[16]. The child becomes an individual through this process. Both male and female children are forced to undergo castration to gain selfhood as assigned by society. This explains the development of masculine and feminine characteristic traits in individuals. The linguistic structure creates masculine and feminine natures, not biology, chromosomes, or genetic laws. Thus, if masculine and feminine natures are fragile, they can be subverted within the linguistic process itself.

Since a society's language determines its conventions, morals, and cultural practices, any reorganisation of the structure of that society requires altering the existing structure of its *langue.* If the meaning of phenomena is arbitrarily decided by the nihilistic, metaphysical, profit-oriented forces, the rereading and reinterpretation of phenomena is the means to liberate meaning from the overarching control of the traditional discourses. As Derrida conceived, meaning will be made plural in linguistic expressions produced by human contact with nature. Language and signs that enable the infinite play of signification would be able to articulate those experiences. In 'writing', language is turned into a force that disrupts all identities to manifest plural senses of our experience of things. As one of the masculine discourses of representation, such writings shall be considered a feminine play of language. The feminine here does not mean the gender-specific female, as some feminist critiques of Derrida attribute[17]. On the other hand, the feminine marks the 'other side' of the legitimate experiences and meanings recognised by the patriarchal-nihilistic civilisation. Though such writings have always been pushed to the margins, they can still produce changes in the unconscious structure of the people by altering their *langue.* The reflections of Saussure and Nietzsche on the nature of the human experience of truth, thus, have brought the issue of truth to its logical consequences in the hands of poststructuralism.

**Funding:** This research received no external funding.

**Institutional Review Board Statement:** Not applicable.

**Informed Consent Statement:** Not applicable.

**Data Availability Statement:** All data are from hard copies of books and journals available in various libraries and from the personal collection of the author. Few soft copies are taken from J-Store routed through my university library.

**Conflicts of Interest:** The author declares no conflict of interest.

## Notes

1. Affects' of the body is a key notion found in the philosophies of Spinoza, Gilles Deleuze, Guattari, Brian Massumi, Sara Ahmed and Lauren Berlant. Sensations from external phenomena are impacted on human body leading to activation of different states, intensities and affections. Positive stimuli from objects such as pleasant smell, colors of seasons, flowers, trees, rainbow, moon, stars, dusk, blue ocean trigger intensities by which the power of a body to act is increased. Positive affects creates joyfulness leading to the release of hormones, whereas negative affects produces sadness and passivity. Affects are the fundamental building blocks of experience. Affects leads to emergence of language, ideas and works of arts like literature, painting and music.

2. This turn in continental philosophy took place under the influence of Ferdinand De Saussure which is similar to the one produced by Gotleb Frege in the analytical philosophy. With the linguistic turn language is no longer considered as a mere devise for representing already existing meaning of the things. Rather, meaning itself is understood as constituted in and through language. This is against the representative function of language explained by metaphysics.

3. Ecriture feminine comes from French which refers to a style of writing that articulates female experiences.

4. Though ancient philosophers like Heraclitus and Buddha had also underlined the chaotic nature of the world, they never employed this truth to confront the philosophies of 'being'.

5. Recent privileging of the body and libido over soul and consciousness by the poststructuralist philosophy is seen as an attempt to regain the value of worldly life, which was depreciated in recent human civilization. Modern societies treat sensuality as harmful to rational goals of mankind. It is observed that philosophy for the last two millenniums were instrumental in denying the sensual-bodily desires as inferior to the transcendental realities of soul and spirit.

6. Helene Cixous evaluates the male authors such as James Joyce and Jean Genet also to be woman writers.

7. Gilles Deleuze and Felix Guattari say bodily intensities are force-fields generated in the body when it is affected by external objects. The sensation of beautiful objects creates intensities in the body by which the power of the body to act is increased (Deleuze and Guattari (2002), *Thousand Plateaus*, pp. 157–158). These intensities turned later into aesthetic images, enabling the production of works of art.

8. Kristeva's semanalytical researches show that the real origin of language is in the bodily desires, which in the case of human infants, exists in semiotic form. However, as the child grows, the semiotic from of language is replaced by symbolic expressions that constitute the structure of the language of the society. But the language of poetry in the opinion of Kristeva recaptures the lost early bodily drives that exist in a suppressed form. Kristeva's proposal for women to write their body is meant to express the world of desires.

9. It is the underlying structure of language consisting of rules, conventions and principles that govern a particular language.

10. Psychoanalysis explains that all human beings carry masculine and feminine drives irrespective of their biological sexual difference.

11. There are two faces for woman in Nietzsche. One is the feminist woman who hates sense and beauty inorder to become like a man. Like the masculine philosopher, she hates the world and dominates over it like men dominate the world through construction of systems of truth. Nietzsche's misogyny is directed against the feminist woman who devaluates body, sensuality and love. Another face of woman in Nietzsche's philosophy is the 'affirmative woman' who accepts life, beauty, body, senses and the world, who is beloved of Nietzsche. Nietzsche opposed the enlightenment of women and men because he perceived it as a project to cut men and women from their natural instincts to make them more rational, conceptual and scientific, thereby turning them away from sense, body, and beautiful appearance of the world. Nietzsche viewed feminism as a masculine project which is against the feminine stance of his philosophy. It is not to argue that the entire corpus of Nietzsche expresses feminine thought. There are also statements in his writings that uphold a masculine approach to the world. We have to take ideas selectively from various texts of Nietzsche that explain how a feminine way of approaching truth can function in philosophy.

12. Nietzsche claims that truth is only our perspective of phenomena.

13. Aristotle in Politics associated the male sex with reason and the female sex with body or emotion. Jean Jacques Rousseau in Emile, associated women with emotion and men with reason and logic. Kant in his Anthropology, positioned men as more capable of rational thought and moral reasoning than women.

14. Nietzsche's critical comments against women are directed against the masculine type of woman, the feminist, because the feminist woman wants to resemble man, the dogmatic philosopher, demanding truth, science, objectivity, and illusions of male virility. He thinks that feminism makes women sick by setting enlightenment as their goal. Instead, he perceives woman as more natural, gracious, and playful than the man (Nietzsche, 1989, [11], p. 163).

15. Heidegger distinguishes ontic nature of metaphysics from fundamental ontology. Ontologically, truth of an entity is its presence appeared before human consciousness in concrete particular instances. Ontic denies the particular experiences of things while categorizing them hierarchically in terms of genera and species.

16. Unlike Freud's idea of castration as a loss of the genital organ of the child, for Lacan it is the loss of early imaginary plenitude of the child resulted by its entry into the symbolic structure of the society, constituted by language.

17. Conventional feminists have criticized Derrida for associating woman's truth with linguistic styles. See Derrida and Feminism: Recasting the Question of Women, Routledge, 1997. To some of them Derrida trivializes the women's oppression in society while looking at it as a question of language.

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
