# Peer review of "Formation of Feminine Truth in Poststructuralism"

_philosophies, doi:10.3390/philosophies8050079_

Round 1

Reviewer 1 Report

This view of truth proposed by Nietzsche was a stimulant to the feminist thinkers of poststructuralism to say that they disown the traditional perspective of truth and perceive the reality of the world and values differently. They ventured out to produce writings that enabled them the expression of their experience of reality. They call this the ecriture feminine, the feminine mode of writing.” Why root this in Nietzsche particularly? I agree that the idea resonates with poststructuralist feminism, but it may not be the sole source of inspiration and these claims here may be unwarranted. This sounds like an overstatement to me & I think it must be rephrased. But if the author would still like to defend it, then there is more work to be done here. Moreover, it seems rather problematic to present Nietzsche, a renowed misogynist, as a primary source of inspiration for poststructuralist feminism--this issue also needs to be addressed/justified.

If femininity and masculinity in this context have nothing to do with men & women or gender identity, why are they called as such? What is the point of calling them feminine and masculine, which inevitably brings to mind women and men, and the binary system of gender?

Nietzsche’s criticism of nihilism can be seen as a direct attack on the masculine perspective of the world.” It seems to me that there is more to say here about why this view is masculine (especially if these words do not correspond to an essentialist notion of gender). Why associate productivity, capitalism, etc. with masculinity? Is it merely due to convention? Yet there is nothing conventional about ecriture feminine--so how come it is unconventional when it is feminine, and conventional when masculine? Since these questions are often left up in the air in Cixous and Irigaray, it seems like the author should find a way to address these issues.

Though Nietzsche does not critique patriarchy in any place of his works, it is observed that his criticism of nihilism of metaphysics was an indirect attack on patriarchy”—But Nietzsche himself saw feminism as a sign of decadence, in line with nihilism. Just as he exalted aristocracy, he was also invested in patriarchy. See: Schotten’s Nietzsche’s Revolution. I think this contests Bergoffen's point--though there are ways to read Nietzsche against himself here (again, this may be an issue of phrasing).

I think this a well-written and insightful essay deserving of publication once these points are addressed.

Abstract- Instead of “it,” should begin with “This essay”

The use of “it” in the beginning of the essay is confusing—the first few sentences need to be rephrased.

Accents in French words are needed: é, è, etc.

Needs to be proofread—some errors.

Author Response

  1. The reviewer cast doubt on presenting Nietzsche as a source of inspiration for poststructural feminist writing due to the existing conception of Nietzsche as a misogynist.  Therefore he asked me to either rephrase my writing or justify my position by adding more writing. I preferred defending my claim by bringing in justification by adding arguments. A detailed explanation has been added to the essay. All the sentences marked with yellow in the essay are my defences for presenting Nietzsche as a pro-feminine thinker
  2. The reviewer asks that if masculinity and femininity in this context have nothing to do with man & woman or gender identity, why are they called as such? I have given arguments to justify this position in the essay and marked it with yellow colour. 
  3. The reviewer asks why the essay associates productivity, capitalism etc., with masculinity. Justification is added in the essay, which is highlighted with yellow colour.
  4. Another point raised by the reviewer is that since Nietzsche saw feminism as a sign of decadence, how can his criticism of nihilism be an indirect criticism of patriarchy?   

         More writings have been added to explain it in a detailed manner. End notes are also provided in support of it. References from other authors have also been brought in to defend the claim.

5.  Sentences in the abstract have been rewritten as suggested by the reviewer.

6. The first few sentences of the beginning of the essay have been rephrased as asked by the reviewer. They are typed in blue colour.

7. French alphabets for the French word 'ecriture feminine' could not be brought in due to my lack of skill in French typewriting.

8. Proofreading of the language of the essay has been done again.

Reviewer 2 Report

Please see the uploaded file

Author Response

The reviewer has not set any conditions to publish the article in the journal. Since the reviewer does not suggest any revision or modification in the essay, I would not like to give any response to the reviewer at this stage.

Reviewer 3 Report

The author attempts to argue that there is a paradigmatic turn that has taken place in the history of philosophical thinking, that is, a turn towards the feminine. The crux of the argument is that Nietzsche's and Saussure's interventions provided the conceptual framework for such a turn. Author argues that the feminine has to be understood as 'the experience of plurality produced in the affectedness of the body by the world' and that the traditional philosophy could not capture this experience of plurality. Various aspects of Nietzsche's philosophy and of Saussure-inspired perspectives have been discussed elaborately in the paper in support of the claim regarding 'the feminine' nature of what the author called the post-metaphysical thought. I would consider this as the major strength of the paper. However, the weakness, to my mind, is that the argument is based on a strong binary between traditional as metaphysical and the Nietzsche/Saussure-inspired contemporary thinking as post-metaphysical, former as masculine and the later as feminine. The author appears to make some sweeping statements such as traditional philosophy was  representational, philosophy before Nietzsche has been essentialist, transcendental and hence masculine, and so on. My suggestion to the author would be to revise those statements in such manner that they are better substantiated. 

Language is fine, and no grammatical issues detected. Minor editing regarding the style of the usages especially of the philosophically loaded terms such as 'transcendental', 'metaphysical', 'essentialist' etc is suggested for making the paper academically more sound. 

Author Response

  1. The reviewer suggests that the author modify the essay's claim that there is a strong binary between traditional philosophy as metaphysical and the Nietzsche/Saussure-inspired contemporary thinking as post-metaphysical. The reviewer seems doubtful about presenting the former as masculine and the latter as feminine.  

Complying with the reviewer's suggestion, I modified the claim. I changed the characterisation of philosophies before Nietzsche and Saussure as metaphysical and masculine and the philosophies after Nietzsche and Saussure as post-metaphysical. 

I provided sufficient justifications/reasons for viewing metaphysical philosophies as masculine.

The essay no longer says that all Nietzsche/Saussure - inspired contemporary thinking are masculine.

2.  The essay sufficiently explains why the language used in metaphysical philosophies serves a representative function.  Saussure's structuralism clarifies the question of how language can function differently.

3. As per the reviewer's suggestion, terms such as metaphysical and transcendental have been reworded wherever possible.

4) changed sentences and words are typed in blue colour. 
